# Who Can Help Us on This Journey? African American Woman with Breast Cancer: Living in a City with Extreme Health Disparities

**DOI:** 10.3390/ijerph17041126

**Published:** 2020-02-11

**Authors:** Shelley White-Means, Jill Dapremont, Barbara D Davis, Tronlyn Thompson

**Affiliations:** 1College of Graduate Health Science, University of Tennessee Health Science Center, 910 Madison, Suite 317, Memphis, TN 38163, USA; 2Loewenberg College of Nursing, University of Memphis, 3567 Community Health Building, Memphis, TN 38152, USA; jdaprmnt@memphis.edu; 3Department of Management, Fogelman College of Business and Economics, University of Memphis, Memphis, TN 38152, USA; bddavis@memphis.edu; 4Department of Biology, College of Arts and Sciences, Howard University, 2251 Sherman Avenue NW, Washington, DC 20001, USA; Tronlyn.Thompson@bison.howard.edu

**Keywords:** health disparities, breast cancer, support agencies, African American women, focus group, race, access, education, emotional support

## Abstract

This qualitative descriptive research study looks at the services that community-based breast cancer support agencies provide to underserved and African American women who are at risk for or diagnosed with breast cancer in Memphis, Tennessee. We seek their understanding of breast cancer mortality disparities in Memphis. Data were collected using semi-structured in-depth focus groups with five breast cancer support agencies. Categories and patterns were established using thematic analysis and a deductive *a priori* template of codes. Thematic analysis is a method for identifying, analyzing, and reporting themes within the data. The main themes identified within support agencies for African American women with breast cancer who live in Memphis were barriers to the use of services, education, health system support, and emotional support. Numerous sub themes included cost of medications, support group supplemental programming, eligibility for mobile services, patient/provider communication, optimism about the future, and family advice. Procrastinating, seeking second options, fearfulness, insurance, childcare, and transportation were barriers to care. Community-based breast cancer support agencies play a critical role as connectors for women with breast cancer who live in medically underserved areas and must find their way within a fragmented medical care system.

## 1. Introduction

Community-based breast cancer support agencies who address non-medical, social determinants of health needs that serve as barriers to maximizing breast health outcomes may play a vital role in mitigating breast cancer mortality. In various ways, they share a common emphasis on addressing social, economic, and psychological needs of breast cancer survivors and those at risk of breast cancer. Services provided by these agencies complement services provided by primary care and oncological treatment providers as well as family caregivers. Community-based breast cancer support agencies include patient navigators (generally affiliated with hospitals or comprehensive clinics), local breast and cervical cancer and early detection programs (funded by CDC), and breast cancer support groups (sometimes referred to as supportive-expressive group therapy). The types of services provided by various community-based breast cancer support agencies are diverse and sometimes overlapping. The supply of these community-based support agencies varies by city, particularly support groups that are founded by and designed to focus on the needs of African American women.

Patient navigators provide services such as education and awareness, which includes identifying and addressing barriers to care, scheduling appointments, attending appointments, and providing or facilitating referrals to medical care and other support agencies [1]. As a result of their vital roles as community contacts for patients who live in medically underserved communities, patient navigators have been documented to increase access to care [2], increase cancer screenings and timely cancer treatment initiations [3,4], increase completion of diagnostic procedures among women who missed follow-up diagnostic appointments [5], and decrease time to diagnosis for women with abnormal breast screenings [6]. 

Another critical community-based breast cancer support agency is the National Breast and Cervical Cancer Early Detection Program (NBCCEDP). The program is a national community health service administered at the state level. A major design of the program is to reduce time between abnormal screening and diagnosis (no more than 60 days) as well as the time between diagnosis and treatment (no more than 60 days) for breast and cervical cancer patients who live in underserved areas. Through the NBCCEDP, low-income, uninsured and underinsured women receive access to free screening and diagnostic services, access to treatment via Medicaid, and access to patient navigation services. How effective has the program been? The NBCCEDP has been effective in reducing time between abnormal screening and diagnosis and overall time between abnormal screening and treatment [7]. However, the success in reducing time to diagnosis and time to treatment varies by race/ethnicity; racial and ethnic minorities have longer time periods than whites, creating disparities among these groups [8,9,10]. The NBCCEDP has also reduced breast cancer mortality [11,12]. 

A third type of community-based breast cancer support agency is the local community breast cancer support group. Sometimes referred to as supportive–expressive group therapy, breast cancer survivors (and in some cases their family caregivers) meet to discuss breast cancer survival strategies, available resources, strategies for communicating with providers, access to community resources, and even death and dying. Consistently, support groups have been found to provide psychologic benefits including reduced depression and mood disturbance [13], reduced perceptions of pain [14], and enhanced survival rates [15].

Community-based breast cancer support agencies have a unique window into the lives of women who strive to maintain their breast health and survive breast cancer. Thus, in this paper, the third in a series of papers seeking to understand why the rate of breast cancer mortality is two times higher for African American women than white women in Memphis [16], we sought to explore this breast mortality challenge from a different vantage point. In the first paper of this series, we sought answers from a focus group session with breast cancer survivors who shared strategies that helped them to survive or hindered them from surviving breast cancer in Memphis [17]. The second paper was based on interviews with oncologists regarding their perceptions of the circumstances (personal, health system, and community related) that caused challenges for their patients to maintain their breast health [18]. The purpose of this third study was to seek insights regarding racial disparities in breast health outcomes in Memphis from key service providers for screening and breast cancer survivorship, in other words, community-based breast cancer support agencies (navigators, the state administered national breast and cervical cancer program, and breast cancer community support groups). As care support agencies that are available in the community for women in Memphis, these service providers have a close-up view of the circumstances and decision-making processes among women at risk of and surviving breast cancer as well as a close view of what occurs in the primary care, surgical, and insurance environments impacting these women. We report how support agency providers perceive the challenges that African American women face in maintaining their breast health as well as potential solutions.

## 2. Methods 

### 2.1. Design and Setting

This focus group study used descriptive qualitative analysis to survey various community-based breast cancer support agency representatives who provide care to underserved African American women who were at risk for or diagnosed as breast cancer patients in Memphis, Tennessee, one of the largest cities in the Mid-South region of the United States. A purposive sampling strategy was used for contact with all the breast cancer support agencies whose primary focus emphasized care for underserved and African American women. Five of six Memphis community-based breast cancer support agencies participated in the focus group interviews conducted by an experienced focus group facilitator. The focus group lasted approximately 90 min. A follow-up interview was conducted with one participant to gather complete information because the representative attended the focus group late. This telephone interview lasted about two hours and was conducted by the project principle investigator. During data collection, participants expressed a variety of emotions including laughing, joking, and even disbelief as they discussed experiences when dealing with underserved and African American women who were at risk of or diagnosed with breast cancer. The focus group and telephone interviews were transcribed by an experienced transcriptionist and reviewed by the interviewer and principle investigator researcher for accuracy of transcription. The rationale for using a focus group was to learn what community-based breast cancer support agencies perceived and described as challenges faced by underserved and African American women at risk or diagnosed with breast cancer. Focus groups yielded information from support agencies about their meetings, interactions, and the services used by the individual dealing with the diagnosis or potential diagnosis of breast cancer. This type of yield, symbolic interaction, was what Patton [19] identified as the strength of this strategy. This research was funded by the Tennessee Department of Health as part of the Association of State and Territorial Health Officials (ASTHO) Breast Cancer Learning Community Project. 

### 2.2. Community-Based Breast Cancer Support Agencies Represented

Participants in the focus group included representatives from three breast cancer support groups, a patient navigator organization, and a Tennessee Breast and Cervical Cancer Program. The services provided by these agencies included monthly support meetings, financial assistance, referrals, meals, access to screenings, medical copays, wigs, bras, prosthetics, assistance in enrolling in TennCare (i.e., Tennessee Medicaid), and thus access to treatment services, referrals to patient advocacy services as well as volunteer breast cancer survivors who are matched with women living with breast cancer in order to provide a “buddy” to meet with the women face-to-face and call on the phone, and patient navigators who work with women once treatment is completed and assist with reconstructive surgery, etc. 

### 2.3. Instruments

A nine-question semi-structured interview guide with sub-questions was developed by the Principal Investigator and two Co-PIs. Questions were piloted for clarity to the topic by three PharmD candidates in their third year of pharmacy school to promote content validity and establish rigor and trustworthiness. Questions were developed using insights from previous work with breast cancer survivors and Co-PI experience with support group participants. 

### 2.4. Procedures: Data Collection

Institutional review board approval was granted for this study from the affiliated university. The researcher solicited community-based breast cancer support agencies providing care to underserved and African American women who were at risk for or diagnosed with breast cancer. At the time of our study, Memphians had local access to the state breast and cervical cancer program, aff three (3) breast cancer support groups emphasizing the needs of African American women, one navigator program affiliated with one of the city’s largest hospitals, and a navigator program affiliated with the American Cancer Society. Among the five participants in our focus group, we had at least one person representing each of the community-based breast cancer support agencies, with the exception of a representative of the navigator program at the hospital.

Support group services agreeing to participate in the study were given the date and location of the focus group session. After each support group representative presented for the focus group, verbal consent was obtained from individuals who agreed to participate in the study. Next, the focus group was conducted with participants who presented to the session. Participants were encouraged to be open and honest and were asked open-ended questions, followed by probing questions when necessary, to reveal details regarding experiences. Nine main questions with sub-questions were asked based upon the researcher developed questionnaire guide.

### 2.5. Data Analysis Strategy

Qualitative data analysis techniques were used to illuminate the data. Open coding of the data began with the two qualitative researchers reading the interview transcripts several times. Next, a qualified qualitative researcher reviewed the transcripts. Coding the data manually enables a rich interpretive analysis. A thematic analysis and deductive a priori template of codes was used to make sense of the data [19,20]. This method produced a structure to handle, organize, and derive meaning from the focus group data. The template for the coding was based on the interview questions and helped to identify overarching themes. The initial template was developed to represent a depiction of the themes identified in the data. The template was applied to each transcript in turn, coding all relevant segments. Data management was performed through development of suitable classification or coding schemes by breaking down the data into discrete parts closely examined, paralleled for similarities and differences, and compared repeatedly until saturation was determined [19,20]. During open coding, common topics within the data from the participants and through different levels of data were identified that described common themes. Internal quality and trustworthiness were attained by having one of the researchers (an author on the paper) independently code the data and form consensus through analysis. The data analysis reported in Table 1 and Table 2 represents themes and sub themes deducted from a focus group of community-based breast cancer support agencies conducted in spring 2018 by a moderator.

## 3. Results

Table 1 reports the demographic characteristics of the support agency representatives who participated in the focus group. All were African American women from local breast cancer support agencies and ranged in age from 54–68. The women represented diverse age ranges, years of employment at the agency, and education attainment. 

Table 2 reports the four major themes that were identified in the focus group: (1) Barriers to use of services; (2) Education; (3) Health system support; and (4) Emotional support. Themes were extracted from support group representative responses to several of the survey questions. A summary of the participants’ responses were organized according to the four major themes. The summaries focus on factors that explain some of the challenges needing to be addressed in order to help African American women survive breast cancer experiences. 

### 3.1. Barriers to Use of Services 

The Tennessee Breast and Cervical Cancer program provides the gateway to access to detection and treatment after screening finds something suspicious. However, women may not access services due to fear, procrastination, desire for a second opinion, unwillingness to accept the diagnosis, or lack of transportation. 

One respondent noted that a woman’s mindset might be her own access barrier. She described a person who knew she had a family history of breast cancer and knew that with treatment she would be able to survive, yet she refused to get a mammogram. The conversation follows: “Oh, my mom had breast cancer’; and “How is she doing?”; “Well, she’s good. It’s been a year ago but she don’t tell me nothing about it ‘cause I don’t have breast cancer and I’m not going to get a mammogram.”

One-to-one conversations with navigators can also make a difference. One respondent indicated that sometimes you need to be persistent and persuasive in talking with women with a suspicious growth or bleeding. One support group participant described the intensity of the support group worker’s involvement as follows: “it took J**** to go and speak with this lady continuously, never fussed, just to go with her, and eventually she did change her mind. Um, but you can’t just tell someone ‘well you need to go.’ What you do is you need to be there with them, and she was there with her, took her there, was there with the doctor, and all of that, and even when the doctor said there isn’t anything we can do about it, and even when she was dying, so it, it takes an individual not to just say a few things and get them to change, stay with them.”

The other way to address fear or a failure to perceive the benefits of screening, diagnosis, and treatment is education and awareness:
“when you’re informed and educate people, they’re better able to go through the process and do what they need to do. When they don’t know, the lack of knowledge creates more fear.”

Out-of-pocket costs are mitigated by the breast and cervical cancer program because once enrolled in the program, screening is free. However, cost of medications may be a barrier, even for women with health insurance coverage.
“We have people—they get these drugs and they’re $900 a pill, you know, what do you do with that? …[insurance] does not cover that $900 medication and if someone in the office does not literally take it upon themselves to call the pharmaceutical company to try to set it [medication assistance program] up, then that’s a great barrier.”


One participant said that the women need help in selecting the Affordable Care Act marketplace plan, so they are better able to have a plan to cover their medications and are not underinsured. However, then another dilemma arises:
“… but then you also have to consider then it as maybe a financial barrier for her to pick a platinum plan that will give her the ultimate benefits that she needs, so therefore she has to look at it ‘can I pay my house note or pay this or do I need to pay this premium that will give me this return on my investment’ so it’s kind of a balancing act for them they have to look at what’s going to be effective ….and if it’s a new patient, they’re not ill, they’re in pretty good health so they’re not looking long down the road, especially for breast cancer and that’s a whole other problem.”

The breast and cervical cancer program has a critical role in this situation:
“… but one way we have been able to get around that issue is that we give them an opportunity to say if you have a plan that does not pay for the services that you need then you’re considered underinsured. So at that point we can enroll them in the Medicaid system, it will be secondary, but it will offset some of that, supplement the cost, in addition to that we can also offer them the opportunity to drop that plan and then we enroll them in the Medicaid program.”

Out-of-pocket payment challenges may occur during any phase of the screening, diagnosis, treatment, and follow-up periods.
“Out of pocket costs: Is most challenging during follow up, and you have no job
If you are in midtown (Knight Arnold), you are okay with getting the busIn 38106, 38109, and 38116, (where it is a 1 hour wait for a bus) you can’t stand in the sun after radiation, and you are nauseous after chemo38115 (Hacks Cross is where the children*** live) and have to go to 38106, 38109, 38116, and then take you back east (STUCK IN POVERTY)” 

Participants noted that access may also be limited because of where certain racial/ethnic and socioeconomic groups live.
“…like 38109 it’s a desert, a medical desert, so women in those zip code areas are a little bit less inclined to seek services because they are not there, they have to go outside the radius of 10 to 20 miles outside of your neighborhood or zip code to get services, it’s a barrier for them.” 

### 3.2. Education

Because women are visual, breast demonstrations and models that reflect the various stages of the cancer are most helpful for their understanding of the impacts of breast cancer and what the outcome may look like. While demonstrations are provided, a demonstration is a good time to “dispel some of the myths and fears they have.”

Support agencies report that a short term educational training provided through a half-day health fair is not enough. Support is better if it is one-on-one and continuous. For example, monthly educational seminars are effective because women are “… always doing something and learning something.” Additionally, training that actively engages cancer survivors is effective. One focus group participant described a skit their group used called Hats Off; the skit is funny and it “… covers every possible reason why our ladies don’t go for their mammograms, why they don’t go to the doctor’s office…[It] does have a deep, deep, deep message, and it gets people to pay attention. They’re laughing a lot, but they hear the message, and that works.” 

### 3.3. Health System Support

As mobile breast screening vans travel to locations and neighborhoods, many think they would enhance access for women living in underserved areas. However, focus group participants indicate that substantial barriers exist in their use, red tape such as requirements that the women have a primary care doctor or a medical home. The breast and cervical cancer program is able to declare that the screening program is their medical home, and then the patient can move directly to screening. Another criterion for the mobile vans that creates a barrier for low income women is the poverty level income criteria. For the mobile vans, the benchmark poverty criterium is that a woman must be less than 100–150% of poverty. This criterium is too low for some low wage working women, who would benefit from the mobile van using the highest federal poverty level screening of 250%. The Tennessee Breast and Cervical Cancer Program uses the 250% of poverty cut-off for screening. 

While transportation is a problem for some breast cancer patients, it is only a moderate challenge if the patients are connected with support agencies. For the later stage patients, transportation to screenings can be arranged by calling the support group and someone will come by to pick them up and transport them. Medicaid pays for transportation for medical appointments. However, a greater challenge, even when support agency assistance is available, is the care of their children. Medicaid will not pay for the transport of children or for childcare. 

Affiliation with the breast and cervical cancer program has provided women the support needed to address the structural barrier of hours of operation that are not convenient for working women. For example, one participant noted, “I know that we have had and partnered with *** and *** [hospitals] to do some sort of midnight mammograms after hours and Saturday appointments….” Barriers in scheduling have been mitigated with the assistance of navigators. 

Sometimes the doctor is the barrier to effective care and breast cancer survival.
“…they don’t have good bedside manners, and then they put even more fear in her when she comes in. Often times they’ll talk and say things the woman don’t understand and so you need to break it down in layman terms and maybe have someone to go with you to that appointment to help you understand what they’re talking about.”

Sometimes, the way care is managed in the physician’s office may become a barrier. This is a health system concern.
“If a lady doesn’t have a good primary care doctor, it might be months before the person sees the doctor; they may not meet until the next scheduled visit, which would be 6 months; may not have a proactive person at the office; the mammogram information could be put in the person’s file and not be seen by the doctor until you get back to your next visit, which could be months later; information can be stuffed in the file and lost among other papers in the file, and missed the next time you see the doctor; also can get lost because the doctor doesn’t find out if you have transportation, childcare, nor expresses the urgency of you getting to mammogram or other diagnosis; lack of follow up by primary care provider, everybody doesn’t get sent all testing information. A patient can get lost in the 3-6 month follow up gap.”


### 3.4. Emotional Support

Back in the day, grandmothers were a barrier to treatment. They would say, “oh honey…you’re fine, if you’re not hurting or anything you’re fine.” Now, women are less likely to get that kind of advice. Sometimes, the appropriate family support is not there because the women don’t want their family members to know that they have cancer. Additionally, some family members may distance themselves if a person has a lot of financial needs all the time, and those family members use their limited resources to take care of themselves.

There are also cultural traditions such as women should not look at their bodies. This particular tradition causes challenges when providers try to get women to perform a breast exam. Then, the challenge becomes a conflict between cultural medicine traditions and western medicine. 

“…so trying to explain to them about a breast exam when your grandmothers told you not to do that, that’s taboo, or use these ointments and all these other kinds of things and you come in and you’ve got full blown breast cancer and you’ve known a year, but you’re trying to treat it yourself because of some remedies you’ve been taught, those are some issues.” 

“…she decided she was going to do all the herbal all of that, and eat right, which worked for 10 years, but she didn’t bring the doctor in with her on the decision, so in ten years I’m getting a call from the pastor, can you help me, I didn’t know the pastor, I didn’t know her, and so when I get to her, I don’t know how the lady stood it, she actually physically had a hole in her side, I watched her, I didn’t change clothes because she showed me, when the doctor came in, it was a younger guy, he had to excuse himself, I watched the color drain out of his whole body, and he called me and said I’m sorry I had to excuse myself, because he never seen anything like that and she had knowledge even when it went to hurting, she still refused, and she was no longer taking care of anybody, she refused to go. She told me the only reason she was going was because she was afraid they were going to smell her because she could smell herself. And I mean that’s horrible in this day and age that you got a norm that’s still hanging on and you can’t get past the taboo.”

Overcoming this problem requires time as well as understanding by the service provider. 

Receiving emotional support from a significant other may affect decision-making. These are some of these possibilities:
“…men in their lives may leave them; husband became mentally abusive once she got breast cancer; men fear doctors and lack knowledge about breast cancer; men don’t realize that women may not have to have a mastectomy, may only have a scar, may not need to have chemo, may live 40–50 years longer.”

## 4. Discussion 

The unique value and contributions of support agency services are reported in this manuscript. Support agency services are so beneficial, but it is unclear whether widespread knowledge of support group services exists in the community. In our on-going online survey of African American women in Memphis, preliminary findings indicate that 24 percent have used the services of the breast and cervical cancer program and 10 percent have used a navigator program. Among African American breast cancer survivors in Memphis, the percentages are 33 percent and 25 percent. One practical implication of these findings is that aggressive and targeted promotion of breast cancer support agencies is needed. While these agency programs are visible during health fairs and during the October breast cancer awareness month, new venues for promotion such as inclusion in year-round health ministry programs at churches and promotion to younger audiences via social media.

Additional intervention strategies gleaned from insights provided by support agencies include educational support opportunities offered on a continuous basis to keep women engaged in breast cancer awareness and treatment, both pre- and post-survival. Support agencies suggest that existing community programs designed to screen and mitigate breast cancer mortality for underserved populations should set realistic poverty benchmark criteria. Furthermore, enhanced efforts to train providers in effective cross-cultural communication are needed. Providers should be aware of cultural norms that may cause women to not be compliant as well as design strategies to challenge such norms. The role of family is critical; there is a need to design programs to educate men about breast cancer and demonstrate how men can assist in facilitating better breast health outcomes for their spouses and significant others. Underserved and impoverished African American women may not have access to high quality providers; are there ways to open doors to high quality care? Finally, it is important to remember the connection between breast cancer treatment and employment outcomes, with African American breast cancer survivors more likely than white survivors to leave labor market employment after breast cancer treatment [21]. The resulting lowered or limited income resources then challenge transportation, medication purchases, and other compliance efforts post-treatment.

Being an insider vs. an outsider to support agency programs can make a difference in terms of whether or not one receives encouragement, up-to-date information about breast cancer and resources, scheduling support, and financial supplements for screening and treatment. Support agencies revealed that they can intervene to curb some of the barriers that they observed for women seeking to maintain their breast health. Many of these barriers have been identified in the literature [17,18,22,23,24,25,26,27,28]. These barriers include (1) the lack of conveniently located screening facilities coupled with mobile mammography utilization criteria that are too stringent; (2) office hours that are in conflict with working hours and employee leave policies; (3) lack the financial resources to maintain compliance with medication regimens; (4) more limited financial resources post-treatment when some women are forced to leave employment prematurely due to disabilities associated with breast cancer; (5) limited health literacy; (6) lack of familial support when facing fears associated with cancer or family cultural traditions that are counter to traditional medicine; and (7) inadequate information about maintaining breast health and depression associated with a perceived lack of ability to maintain health. By identifying themselves as the medical home and implementing less stringent poverty guidelines, support agencies have been able to increase access to the use of mobile mammography services. Support agencies have collaborated with hospitals to provide midnight mammography services. Affordable access to expensive medications is provided when support agencies make contacts that open opportunities to pharmacy benefit programs. Support agencies who are in regular contact with breast cancer survivors post-treatment have been able to acquire immediate information about transportation needs for survivors who are no longer able to work and have limited financial resources to pay for transportation to treatment and have developed transportation strategies for them. Given the complexity of health insurance applications for those with limited health literacy, support agencies assist with enrollment in health insurance programs. They also provide one-on-one personal support that may counter a lack of family support due to cultural norms that contrast traditional medicine norms and provide continuous (monthly) education and active engagement that is designed specifically for the particular issues and challenges of the breast cancer survivor. Community-based breast cancer support agencies play a critical role as connectors for women with breast cancer who live in medically underserved areas and must find their way within a fragmented medical care system.

Oncologists in Memphis identified unique Memphis-specific barriers for African American women to survive breast cancer [17] including computer literacy, availability of support systems during office consultations, and the providers’ limited knowledge about the available community support services. Two of these barriers (support during consultations and limited provider knowledge) could be addressed by greater interaction between community-based breast cancer support agencies and providers. For example, one strategy might be conferences or workshops to inform providers of ways to connect their patients with support agencies and/or facilitate the design of community/provider collaboration strategies. However, it is unclear whether Memphis’ limited supply of community-based breast cancer support agencies would be able to expand their capacity to assist providers without having access to greater financial resources. 

In contrast to the literature, support agencies noted that one barrier that African American women who live in underserved areas face in maintaining breast health is poor physician’s office management. Quite revealing, they noted that in fragmented health care systems, information and patients can be lost to follow-up. This is a health system concern that community-based breast cancer support groups report, but have limited ability to impact. 

## 5. Conclusions

Despite their vital role in reducing breast cancer mortality, community-based breast cancer support agencies are underfunded. They are largely volunteer workers with their feet-on-the-ground who can easily identify needs, but face great difficulty in identifying adequate financial resources in meeting those needs. There is a clear and substantial return on investment and cost savings opportunity by directing more resources to programs of community-based breast cancer support agencies.

## Figures and Tables

**Table 1 ijerph-17-01126-t001:** Demographics of support group representatives.

Demographics	
Number of Participants	5
Gender	Female
Age Range	54–68
Ethnicity	African American
Education Attainment	1 Associates Degree, 1 BS, 2 MA, 1 PhD
Collective years at agencies	100

**Table 2 ijerph-17-01126-t002:** Four major themes of African American breast cancer for the Consortium on Health Education Economic Empowerment and Research (CHEER) focus group.

Theme ^#^	Theme	Sub Themes (with Applicable Quotes)	Quotes Related to Sub Themes
1	Barriers to Use of services	1.1 Cost of Medication 1.2 Refusal to use services	1.1 “People they get these drugs and they’re $900 a pill, you know, what do you do with that?” 1.2 “…don’t tell me nothing about it [Because] I don’t have breast cancer and I’m not going to get mammogram.”
2	Education	2.1 Breast models 2.2 Support group supplemental programming	2.1 “We have breast demonstrations and models…to show them the various stages of what could happen… show them about mammograms” 2.2.1 “We do a Hats Off...like a skit where you have humor…it gets people to pay attention” 2.2.2 “[Support group leader] has someone to come in and to educate on Saturdays so we are always doing something and learning something.”
3	Health System support	3.1 Mobile services 3.2 Healthcare personnel communication	3.1 “Mobile units have too any barriers or stipulations to receive the service that makes it ineffective.” 3.2 “…the doctors come in and they can really turn the patient off because of their etiquette and their style and they don’t have good bedside manners.”
4	EmotionalSupport	4.1 Pitch and tone of words 4.2 Optimism about the future 4.3 Family advice	4.1 “…It’s the way you talk to people though, it’s the way you talk to them, what you say to them, you know, don’t do a lot of fussing…” 4.2 “…so when they actually see the women that do have their hair, they look good…that makes them feel good…” 4.3 “...the grandmothers would say ‘oh honey…you’re fine, if you’re not hurting or anything you’re fine”

Note: Column three reports sub themes that were numbered to correspond with the primary theme, with distinctions in unique individual responses noted by numbers after the decimal point. In column four, example quotes are numbered according to the associated sub theme, with the number after the second decimal denoting unique individual responses.

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
