# Peer review of "Who Can Help Us on This Journey? African American Woman with Breast Cancer: Living in a City with Extreme Health Disparities"

_ijerph, 2020, doi:10.3390/ijerph17041126_

Round 1

Reviewer 1 Report

This paper investigates an important issue for breast cancer disparities in Memphis. A few recommendations for clarifying the paper are listed below:

Abstract: Since this study did not directly look at the impact of support groups on breast cancer mortality, temper language in the final sentence of the abstract.

Introduction (minor): support groups are discussed in paragraph 1 and again in paragraph 3, please make it clear how these two concepts are distinct or group the content together.

Introduction (minor):community support services, support groups, breast cancer support groups and patient navigation are important terms that are used throughout the paper. Please define them and how they relate to one another in the introduction. Use terminology consistently to help guide the reader.

Methods (minor): the authors refer to interviews a couple of times (line 96) when I think they are actually talking about the focus groups. Please clarify.

Methods: What qualitative software was used?

Results (minor): Lines 164-169 were confusing to me as written (though the table was quite intuitive). Please clarify. Consider putting detailed information about interpretation of the table in a footnote for Table 1.

Results: To what extent did focus group members' opinions converge or differ? A few lines on this would be helpful as you discuss Table 1.

Discussion (major): In the last paragraph of the discussion section, please make the link between results and the concluding remarks more clear (I think this will prevent the reader from feeling like new information is being provided and rather that the authors are just expanding upon the findings).

Discussion (major): Please describe how these findings relate to the broader literature on barriers to breast cancer prevention and care. Much work has been done in this area. Describe how your findings expand this work and where your findings are the same or differ. Include the needed references.

Intro/Discussion (major): The objective of the paper as stated in the intro is to understand women's challenges with maintaining breast health (line 79-80)--please make sure the discussion and conclusions align with this objective (e.g., XYZ are the remaining barriers) or clarify the objective as needed.

Reviewer 2 Report

The paper describes the design and analysis of one focus group with five participants describing the understanding that providers of breast cancer support services have of breast cancer mortality disparities in Memphis. The paper is flawed in a number of significant ways:

a mixed population of different sorts of service providers: patient navigators, support group providers and screening program staff. These are three very different kinds of service provision that cannot necessarily be seen as uniform. The paper also refers to community health workers, yet another type of service provider, but none of these are included in the focus group. one focus group with 5 members is too small a sample size from which to draw any conclusions the authors do not address saturation of themes, which is the usual indicator for having reached an appropriate sample size the themes described here have been well-described in previous literature (see Oncologist 201318(9), 986–993; Semin Oncol Nurs, 2015 31(2), 170–177; J Health Dispar Res Pract. 2018;11(1):160–178; Cancer. 2016;122(14):2138–2149)and do not add significantly to our knowledge. 

The use of the phrase "support group/services" throughout the paper is problematic and mixes different models of care, including patient navigation, community health workers, supportive/expressive group support and education/information-sharing. 

The statement "The rationale for using a focus group was to learn what African American breast cancer survivors perceived and described
in their experiences as underserved and African American women at risk or diagnosed with breast cancer." seems to refer to their previous study with African American breast cancer survivors.

I had some difficulty with the thematic analysis. Refusal to use services is not a subtheme of access to use services but rather indicative of a barrier other than access to services. Similarly, breast self-exam is listed as a subtheme of educational material but is more accurately spreading myths as breast self-exams have been shown not to impact outcomes. Physician communication is also a well-known barrier, but is not a structural barrier. Pitch and tone of words is not how I would label "…It’s the way you talk to people though, it’s the way you talk to them, what you say to them,
you know, don’t do a lot of fussing…”. In the same theme, I do not understand the term "public glaring."

I suggest the authors conduct a more rigorous literature review and consider what they can add to the literature. If focus groups of community support providers is the focus, I suggest dividing the providers into different types and conducting more groups with more participants until saturation is reached.

Round 2

Reviewer 2 Report

The manuscript is significantly improved